# Histone Mono-Ubiquitination in Transcriptional Regulation and Its Mark on Life: Emerging Roles in Tissue Development and Disease

**DOI:** 10.3390/cells11152404

**Published:** 2022-08-04

**Authors:** Liat Oss-Ronen, Tzlil Sarusi, Idan Cohen

**Affiliations:** The Shraga Segal Department of Microbiology, Immunology and Genetics, Faculty of Health Science, Ben-Gurion University of the Negev, Beer Sheva 84105, Israel

**Keywords:** histones, mono-ubiquitination, transcription, repression, elongation, Polycomb repressive complex 1, H2AK119ub, RNF20, RNF40, H2BK120ub, development, cancer

## Abstract

Epigenetic regulation plays an essential role in driving precise transcriptional programs during development and homeostasis. Among epigenetic mechanisms, histone mono-ubiquitination has emerged as an important post-transcriptional modification. Two major histone mono-ubiquitination events are the mono-ubiquitination of histone H2A at lysine 119 (H2AK119ub), placed by Polycomb repressive complex 1 (PRC1), and histone H2B lysine 120 mono-ubiquitination (H2BK120ub), placed by the heteromeric RNF20/RNF40 complex. Both of these events play fundamental roles in shaping the chromatin epigenetic landscape and cellular identity. In this review we summarize the current understandings of molecular concepts behind histone mono-ubiquitination, focusing on their recently identified roles in tissue development and pathologies.

## 1. Introduction

The mammalian genome is organized and packed as a highly dynamic nucleoprotein complex–chromatin, whose architecture is constantly altered by epigenetic regulators in response to various cues [1,2,3,4]. The basic unit of the chromatin is the nucleosome, where DNA is associated and wrapped around a histone octamer that encompass two copies of histones H2A, H2B, H3, and H4 [5,6]. The tails of the heteromeric histones stick out from the nucleosome core and can undergo a wide range of post-transcriptional modifications (PTMs), including methylation, acetylation, phosphorylation, and ubiquitination [7,8]. Histone PTMs affect the chromatin structure and accessibility, and thus play important roles in key biological processes such as gene expression and DNA repair [9]. Here, we will specifically focus on the role of histone mono-ubiquitination in transcriptional control.

Histone mono-ubiquitination involves the covalent attachment of ubiquitin, a highly conserved 76 amino acid polypeptide, to a lysine residue in the modified histone [10,11]. This occurs through sequential enzymatic reactions: first, a ubiquitin-activating enzyme (E1) activates ubiquitin in an ATP-dependent manner; then a cystine residue on a ubiquitin-conjugating enzyme (E2) is conjugated to the ubiquitin via a thioester bond; and finally, RING finger ubiquitin-ligases (E3) transfer the ubiquitin from E2 to a target lysine residue in the histone substrate [12,13,14,15,16,17] (Figure 1A). The E3 ubiquitin ligases determine histone substrate recognition and the catalysis of its ubiquitination [18,19]. A number of RING finger proteins involved in various cellular processes have been identified as E3 ligases that modify histone substrates (Figure 1). These include the BRCA1 tumor suppressor, as well as CUL4/B and RNF168, which are all involved in DNA damage response (DDR) [20,21,22,23,24,25,26]. Other E3 ligases such as the RING1A/B members of the Polycomb repressive complex 1 (PRC1), RNF20/40, and NEDD4 are involved in transcriptional regulation [27,28,29,30,31,32,33,34], while UHRF1-mediated histone mono-ubiquitination regulates chromatin structure and accessibility by promoting DNA methylation [35,36]. In addition, several E3 ligases, such as RNF8 that is involved in DDR, promote poly-ubiquitination rather than mono-ubiquitination of histone substrates [37,38,39]. Several histone lysine residues in the mammalian genome have been identified as substrates for histone ubiquitination, and their modifications play a role in various processes including chromatin regulation, DNA repair, and transcription [40,41,42]. The most prevalent ubiquitinated histones are H2A and H2B, with an estimated occurrence rate of up to 15% and 1–2%, respectively, although H3 and H4 ubiquitination also occurs [41,42,43,44,45,46,47] (Figure 1B).

Similar to other histone PTMs, histone ubiquitination is a dynamic and reversible process. The ubiquitin modification can be removed by a group of specialized deubiquitinating enzymes (DUBs), that hydrolyze the peptide bond between the ubiquitin and its histone substrate [48,49]. Interestingly, while each DUB can recognize a specific histone ubiquitination modification, in some cases there are several different DUBs that overlap in the ubiquitinated histone substrate [42,46], suggesting context-dependent functions. Histone ubiquitination crosstalk with other PTMs and specific “readers” of the histone modification [50,51,52,53,54,55,56,57,58,59], thus the addition or removal of ubiquitin on histone tails affects the epigenetic landscape and can alter local chromatin accessibility, DDR related processes, or gene expression.

Among the various histone mono-ubiquitination modifications, two of the most well-studied events that regulate transcription are the mono-ubiquitination of histone H2A at lysine 119 (H2AK119ub) by the PRC1 complex and the mono-ubiquitination of histone H2B at lysine 120 (H2BK120ub) by the RNF20/RNF40 complex [28,29,31,34,60,61,62,63]. Interestingly, these two separate histone mono-ubiquitination events have contrasting impacts on transcriptional regulation, emphasizing the functional complexity behind histone PTMs. In this review we will discuss the current understanding on how histone-specific mono-ubiquitination by PRC1 and RNF20/RNF40 complexes regulates transcription, focusing on their emerging roles in tissue development and disease.

## 2. Roles of Histone Mono-Ubiquitination in Transcriptional Regulation

### 2.1. PRC1-Mediated H2AK119 Mono-Ubiquitination and Transcriptional Repression

The deposition of H2AK119 mono-ubiquitination is catalyzed by PRC1—a multimeric complex that functions in concert with the PRC2 complex and together represent one of the most studied evolutionarily conserved epigenetic mechanisms for gene silencing [64,65,66]. PRC1 and PRC2 largely overlap in their genomic binding, and their associated histone modifications, H2AK119ub and H3K27me3, are enriched in the promoter regions of silent developmental genes [66,67,68,69]. PRC1 is a highly diverse complex that consists of several sub-complexes, each containing a RING finger E3 ubiquitin ligase, either RING1A or RING1B, which is responsible for the catalysis of H2AK119ub [28,31,60]. In mammalian systems, PRC1 sub-complexes are defined by the specific Polycomb group RING finger 1 to 6 (PCGF1–6) subunit that composes the complex core together with RING1A or RING1B [60,70]. A further classification of PRC1 complexes can be made into canonical PRC1 (cPRC1) and non-canonical PRC1 (ncPRC1) complexes. cPRC1 complexes contain in their core PCGF2 (also known as MEL18) or PCGF4 (also known as BMI1) and a Chromobox (CBX) protein (CBX2, 4, 6, 7, or 8) that can recognize PRC2-mediated histone H3 lysine 27 tri-methylation (H3K27me3) and provides a molecular link between PRC1 and PRC2 [70,71,72,73,74]. ncPRC1 complexes lack the CBX subunit, and contain in their core RYBP or YAF2 and one of the PCGF1–6 proteins (ncPRC1.1–ncPRC1.6) [70] (Figure 2A). The recruitment of ncPRC1 complexes is not dependent on PRC2/H3K27me3, and instead can be mediated by the KDM2B subunit of ncPRC1.1 in a generic fashion to non-methylated DNA sequences enriched for CpG islands [75,76,77,78,79], or can be directed to sequence-specific loci via the interaction of ncPRC1 complexes with accessory transcription factors [80,81,82,83]. While PRC1 complexes are all capable of catalyzing H2AK119ub, cPRC1 complexes possess a weaker catalytic activity when compared to ncPRC1 complexes, where the RYBP/YAF2 subunit further stimulates the E3 ubiquitin ligase activity of RING1A/B [70,84,85,86,87]. Furthermore, RYBP possesses a ubiquitin binding domain (UBD) and acts as a reader of H2AK119ub, forming a positive feedback mechanism that supports the spreading of PRC1 and H2AK119ub to adjacent nucleosomes [55,87,88]. Within ncPRC1 complexes, evidence from studies in embryonic stem cells (ESCs) and epidermal progenitors suggests that the majority of H2AK119ub deposition is performed by PCGF3/5-containing ncPRC1 complexes [89,90,91]. However, additional experiments in other tissues and cell types are necessary in order to determine how generic these observations are.

The deposition of H2AK119ub on histone H2A tails is a reversible process. H2AK119ub-specific DUBs, ubiquitin specific peptidase 16 (USP16) and 2A-DUB (also known as MYSM1), were shown to be associated with promoters of active genes and to counteract the deposition of H2AK119ub [92,93]. H2AK119ub is also being erased by the BRCA1 associated protein 1 (BAP1), which forms together with ASXL proteins the Polycomb repressive deubiquitinase complex (PR-DUB) [94]. On the one hand, BAP1 constrains H2AK119ub deposition and increases PRC1 stability at target genes by preventing diffused accumulation of H2AK119ub at atypical genomic loci. Yet, on the other hand, the loss of BAP1 in mouse ESCs leads to spreading and genome-wide accumulation of H2AK119ub, resulting in reductions in global gene expression [95,96]. The recruitment of ZRF1, which acts as a reader of H2AK119ub via its UBD motif, also counteracts PRC1 activity and promotes the displacement of PRC1 [97,98]. Altogether, these examples demonstrate the existence of various mechanisms that regulate the deposition and removal of the H2AK119ub mark.

Although it is well-known that the H2AK119ub histone modification regulates transcriptional repression, the mechanisms of this regulation are not fully understood. In part, H2AK119ub regulates transcription by promoting the recruitment of PRC2 to chromatin through interactions with the PRC2 subunit JARID2. JARID2 can recognize H2AK119ub-containing nucleosomes via its UBD motif, leading to the deposition of PRC2-mediated H3K27me3 on H2AK119ub-marked genomic loci (Figure 2A) [54,55,99,100]. In turn, cPRC1 complexes are recruited, synergize with PRC2, and play important roles in shaping the 3D organization of the chromatin [64,101,102,103,104]. Specifically, the polymerization activity of cPRC1 Polyhomeotic homologue (PHC) proteins promote sub-nuclear clustering of PRC1, in which high-order chromatin organization and long-range interactions between Polycomb-bound loci are established and contribute to chromatin compaction—a signature of Polycomb-mediated transcriptional repression [105,106,107]. Such observations have questioned the functional importance of the H2AK119ub histone modification for transcriptional repression per se, which remained controversial up until recently [60]. However, several studies using catalytic-inactive PRC1 mutant cells have shown that the catalysis of H2AK119ub plays a central role in transcriptional repression, and that the loss of PRC1 catalysis results in similar defects in gene repression as seen in PRC1-null cells [90,108]. Mechanistically, H2AK119ub may repress gene expression by interfering with the transcriptional machinery during its assembly by RNA polymerase II and the pre-initiation complex [52,109,110,111], or by blocking the release of poised RNA polymerase II from promoter regions and thus affecting the transcriptional elongation process (Figure 2B) [112,113,114]. In part, this can be indirectly through the stabilization of PRC1 and PRC2 complexes binding at silent genes, due to their ability to act as readers of the H2AK119ub modification [54,55,88,99,100]. Further genomic and biochemical studies focused on the direct interactions between H2AK119ub and the transcriptional machinery are required in order to determine the exact mechanisms by which H2AK119ub impacts transcription.

### 2.2. The Action Mechanism of RNF20/RNF40 Complex during Transcription

RNF20 and RNF40 are two RING finger E3 ubiquitin ligases that cooperate to form a heteromeric complex that catalyzes H2BK120 mono-ubiquitination. While RNF20 and RNF40 are paralogues that share amino acid identity of 72% and a similarity of 86% [115], both seem to be obligatory for complex formation and the catalysis of H2BK120ub. The deletion or silencing of either RNF20 or RNF40 decreases global H2BK120ub levels, suggesting that both complexes are required for the catalysis of H2BK120ub in a non-redundant manner [116,117], although it should be noted that each of these E3 ligases is capable of forming a functional homodimer that catalyzes H2BK120ub in vitro [118].

RNF20 and RNF40 are tightly linked to the transcriptional machinery (Figure 3), and the deposition of H2BK120ub typically correlates with transcriptional elongation and high levels of gene expression [30,34,119,120]. Accordingly, H2BK120ub is enriched in gene bodies of transcribed genes, and overlaps with additional marks of active genes such as H3K79me3 [116,121,122,123]. The activity of the RNF20/40 complex depends on specific E2 enzymes, either UBE2A (also known as RAD6A), UBE2B (also known as RAD6B), or UBE2E1 (also known as UbcH6), that interact with RNF20/40 and provide it with the ubiquitin. Several additional proteins also interact with RNF20/40 to form a functional complex that stimulates the catalysis of H2BK120ub by RNF20/40 (Figure 3A). Specifically, RNF20/40 and an E2 ubiquitin conjugating enzyme have been shown to form a complex together with the RNA polymerase associated factor 1 (PAF1) complex, which plays a role in transcriptional elongation [30,34,124]. In a cascade mediated by the Cyclin-dependent kinase 9 (CDK9) subunit of the positive transcription elongation factor b (pTEFb) heterodimer, PAF1 complex is recruited and regulates RNF20/40-mediated H2BK120ub, which facilitates transcription elongation through CDK9-dependent phosphorylation of RNA polymerase II at its carboxyl-terminal domain (CTD) [32,34,125,126,127]. Overexpression of RNF20 or CDK9 results in increased levels of H2BK120ub and gene expression, while the deletion or silencing of RNF20, CDK9, or PAF1 complex components, results in reduced H2BK120ub levels and downregulation of gene expression [32,34,125,128].

WW domain-containing adaptor with coiled-coil (WAC) is another functional partner of RNF20/40 that is required for its E3 ligase activity. WAC directly interacts with RNF20/40 and provides a molecular link between the RNF20/40 complex and RNA polymerase II, through the binding of WAC to the phosphorylated serine 2 residue of RNA polymerase II CTD domain [129]. Similar to CDK9 and the PAF1 complex subunits, the activity of WAC is required for RNF20/40-mediated catalysis of H2BK120ub [129].

The deposition of H2BK120ub is also involved in epigenetic crosstalk with histone methyltransferase complexes that regulate gene expression. Specifically, H2BK120ub stimulates the catalytic activity of disruptor of telomere silencing 1 (DOTL1) towards the methylation of histone H3 at lysine 79 (H3K79me), which is associated with the transcriptional machinery and transcriptional elongation [51,130,131,132,133]. The depletion of RNF20 leads to a reduction in DOTL1-mediated H3K79me3, while the overexpression of RNF20 or the in vitro use of nucleosomes bearing H2BK120ub stimulates DOTL1 catalytic activity [34,130,134]. Similarly, H2BK120ub was also shown to stimulate the catalytic activity of the complex of proteins associated with SET1 (COMPASS) towards the methylation of histone H3 at lysine 4 (H3K4me), which is associated with active transcription [57,135,136,137]. These examples highlight the connection between H2BK120ub deposition and transcription; however, as with other histone PTMs, the H2ABK120ub modification is dynamic and can be removed by specific DUBs. The Spt-Ada-Gcn5 acetyltransferase (SAGA) complex, which contains a DUB module via its USP22 subunit (or Ubp8 in yeast), is required for the removal of H2BK120ub and transcription of SAGA-dependent genes [138,139,140]. Loss of Ubp8 or the SAGA DUB module component ATXN7L3 increases global H2BK123 or H2BK120ub levels, respectively, while silencing of USP22 did not change global levels of H2BK120ub [122,139,140,141,142,143], suggesting that other DUBs may control the levels of H2BK120ub. In line, two USP22 homologous, USP27X and USP51, were recently shown to interact with ATXN7L3 to form complexes not associated with SAGA, which function as H2BK120ub DUBs [144]. However, further analysis is required in order to determine the differences between these DUBs and whether they act in a context dependent manner.

Interestingly, the DUB module of the SAGA complex regulates H2BK120ub both in the context of nucleosomes and in disassembled H2A-H2B dimers that are associated with the facilitates chromatin transcription (FACT) complex [138], indicating that the involvement of H2BK120ub in several different stages of transcription. Indeed, RNF20/40-mediated H2BK120ub is also associated with the activity of the FACT complex in nucleosome disassembly and reassembly during transcription (Figure 3B). H2BK120ub promotes the accumulation of the FACT subunit SPT16, which in turn disassembles histone H2A-H2B dimer from the core nucleosome, thereby removing the nucleosome barrier and enabling RNA polymerase II to pass along the gene body and perform transcription [30,145,146,147,148]. Interestingly, SPT16 also cooperates with H2BK120ub to promote nucleosome reassembly and stability following RNA polymerase II elongation [30,145]. Thus, although H2BK120ub is generally associated with active transcription, these somewhat opposing functions of H2BK120ub on nucleosome dynamics can lead to gene activation or repression in a context-dependent manner [149]. One such potential mechanism by which the deposition of H2BK120ub can exert suppressive function has been reported to occur on oncogenes, where H2BK120ub was shown to interfere with the binding of transcriptional elongation factors and impede RNA polymerase II transcription elongation [150,151]. In addition, genome-wide studies of H2BK123ub deposition in yeast (H2BK120ub in mammals) demonstrated that suppressive H2BK123ub deposition is enriched at quiescent promoters, whereas ubiquitination at active gene bodies promotes transcription [152].

## 3. Roles of PRC1-Mediated H2AK119ub in Tissue Development and Disease

The PRC1 complex have been shown to play critical roles in stem cells and tissue development, starting from its requirement for the progression past the gastrulation stage to the control of proper spatiotemporal tissue patterning during development [153,154,155,156]. PRC1 activity also plays important roles in maintenance of the transcriptional identity of adult stem cells to maintain tissue homeostasis and is often implicated in different types of cancer [153,155,157,158,159]. While the catalysis of H2AK119ub is a hallmark of PRC1 activity [60], it is becoming increasingly clear that, in addition to canonical enzymatic functions that promote histone PTMs, many chromatin regulators, including PRC1, also possess non-enzymatic roles involved in transcriptional control [160]. Indeed, in addition to the catalysis of H2AK119ub, cPRC1 complexes can exert gene repression by promoting phase separation [161,162,163] and high-order chromatin organization and nuclear sub-clustering [102,103,164,165,166], leading to a compacted and transcriptionally silent chromatin. In addition, ncPRC1 complexes are enigmatically localized to active genes, where, in some cases, their localization promotes gene expression [90,167,168,169,170,171,172]. Here, we will specifically discuss recent discoveries from studies focused on the importance of H2AK119ub for mammalian tissue development and homeostasis (Table 1).

### 3.1. PRC1-Mediated H2AK119ub Plays Essential Roles in Tissue Development

The deposition of H2AK119ub can be detected as early as the process of oogenesis as shown in growing oocytes of 7-days old mouse females, with increased levels in fully grown oocytes [175]. The deposition of H2AK119ub in early embryogenesis is highly dynamic. During fertilization and preimplantation stages, maternally inherited H2AK119ub at canonical Polycomb target genes (developmental genes enriched for PRC1/2 binding in mouse ESCs) is being equalized and reaches to similar high levels in the paternal allele at the two-cell stage, while being depleted from atypical targets and gene deserts [175,176]. Moreover, genetic deletion of ncPRC1 subunits PCGF1 and PCGF6 in mouse oocytes causes a reduction in H2AK119ub and results in lower fetus implantation rates and impaired imprinting associated with enlarged placenta [175]. Interestingly, overexpression of PR-DUB [94,177] in zygotes rapidly depletes H2AK119ub levels and results in arrested development at the 4-cell stage [176]. Together, these studies highlight the essential role of PRC1 and H2AK119ub in early embryonic development.

PRC1-mediated H2AK119ub also plays context-dependent roles in development. Using catalytic-inactive *Ring1b* mutants, Illingworth and colleagues demonstrated that the deposition of H2AK119ub by the RING1B subunit of PRC1 is essential for late but not early mouse development [173]. While complete germline deletion of *Ring1b* results in early embryonic lethality before gastrulation [178], catalytic-inactive *Ring1b* mutants progress past gastrulation. In line, transcriptional analysis in catalytic-inactive *Ring1b* mutant ESCs demonstrated that the majority of genes upregulated in *Ring1b*-null ESCs remained silent when only RING1B catalysis was impaired [173]. However, although catalytic-inactive *Ring1b* mutant mice progressed past gastrulation, later during development mutant mice showed developmental defects and prenatal lethality [173]. Similar observations were also made in *drosophila*, where the repression of homeotic genes normally upregulated upon the loss of PRC1 global activity was maintained in PRC1 catalytic inactive mutant flies during early development, but resulted in late developmental defects and lethality [179]. It is worth mentioning that deletion of *Ring1b* catalytic activity in a *Ring1a*-null ESCs background, was shown to result in a massive upregulation of PRC1 target genes, similar to the upregulation of PRC1 target genes observed in *Ring1a/b*-null ECSs [108]. Thus, differential requirement for H2AK119ub levels during development may exist and potentially can be partially compensated through the activity of RING1A.

A tissue-specific requirement for PRC1 catalysis and H2AK119ub in somatic stem cell differentiation was recently reported in the developing skin epithelium. On the molecular level, the deletion of PRC1 catalysis in embryonic epidermal progenitors resulted in a rapid loss of H2AK119ub and a broad upregulation of canonical Polycomb target genes [90], indicating the importance of H2AK119ub for transcriptional repression. At the cellular levels, the loss of H2AK119ub in embryonic epidermal progenitors enables cellular plasticity and resulted in ectopic formation of Merkel cells [90]—a specialized sensory cells of epidermal origin essential for fine-touch sensations [180]. Importantly, the loss of PRC1 catalysis and H2AK119ub in the developing skin of PRC1 mutants recapitulates the main phenotypic alterations of mechanosensory Merkel cell expansion observed upon global deletion of PRC2 activity [181,182], further emphasizing the central role of H2AK119ub in controlling Polycomb-mediated transcriptional regulation.

### 3.2. Emerging Roles for H2AK119ub in Somatic Stem Cells and Tissue Homeostasis

In adult stem cells, epigenetic regulators are thought to play fundamental roles in facilitating changes in gene expression to maintain tissue homeostasis in response to internal and external cues [2,183,184,185]. While the specific role of H2AK119ub in adult stem cells is understudied, a recent study sheds light into the importance of H2AK119ub in tissue homeostasis and response to ultra-violet (UV) exposure. Li et al., showed that the deletion of PRC1 catalysis (i.e., H2AK119ub) in adult murine skin epidermis led to the activation and migration of melanocytes into the skin epidermis and results in atypical and dramatically increased skin pigmentation [186]. The deletion of global PRC1 or PRC2 activity in murine skin epidermis results with similar increase in epidermal pigmentation, suggesting that this phenotype involves canonical Polycomb repressive functions.

Increased epidermal pigmentation is a process that occurs upon exposure to UV, where epidermal cells become pigmented by melanocytes to prevent DNA damage that can lead to skin cancer [187,188]. Interestingly, exposure of mouse skin to low-dose UV radiation leads to the downregulation of the PRC1 subunit RING1B and its associated H2AK119ub [174], suggesting that dynamic deposition of H2AK119ub in the skin may be involved in UV-mediated signaling and skin cancer. Although the role of H2AK119ub in cutaneous melanoma is yet to be determined, the progression of uveal melanoma—a rare and deadly form of melanoma located at the ocular region [186], was recently reported to be driven by loss of PRC1 activity. Specifically, high-risk uveal melanoma cells exhibit low levels of H2AK119ub with nearly complete loss at PRC1 target loci when compared to low-risk uveal melanoma cells [189]. Moreover, pharmacological inhibition of PRC1 catalysis in low-risk uveal melanoma resulted in acquisition of features related to high-risk cells, including the upregulation of genes associated with high-risk and poor prognosis [189]. Taken together, these studies indicate that H2AK119ub plays a role in modulating the behavior of pigment cells. Further dissection of H2AK119ub roles in somatic stem cells is expected to provide much needed insights into the epigenetic regulatory mechanisms controlling tissue homeostasis and disease.

## 4. The Importance of RNF20/40 and H2BK120ub in Tissue Development and Disease

The activity of RNF20/40 and H2BK120ub have been reported to control various cellular processes, including differentiation, proliferation, survival, DNA repair, and inflammatory responses [117,121,190,191,192,193]. In addition, either the gain or loss of RNF20/40 expression is associated with human malignancy [47,149,194], altogether suggesting a central role for H2BK120ub in tissue development and disease. Indeed, in the mammalian system the importance of RNF20/40 is evident from genetic studies in which a germline deletion of *Rnf20* was embryonic lethal [193], hence heterozygous mice or conditional deletions were used for in vivo studies. Here, we summarize the recent findings regarding the in vivo roles of RNF20 and RNF40 related to tissue development and disease (Table 2).

### 4.1. Roles of RNF20 and RNF40 in Embryonic Stem Cells and Tissue Development

The activity of RNF20/40 and H2BK120ub plays a role in various stages of embryonic development, from spermatogenesis to lineage-specific differentiation processes. RNF20 and H2BK120ub levels are high in precursor spermatocytes, but low in round spermatids and mature sperms [195], suggesting a stage-specific role in this process. Indeed, cell-specific genetic deletion of *Rnf20* in mouse germ cells resulted in reduced testis size, impaired meiotic recombination, and infertility. Interestingly, chromatin relaxation using chloroquine partially rescued the impaired spermatogenesis process in *Rnf20*-deficient spermatocytes, resulting in meiotic recombination and increased numbers of mature sperm cells [195]. Likewise, stage-specific role for H2BK120ub was also reported in ESCs, where H2BK120ub is increased substantially in early stages of differentiation and was shown to be required for the differentiation of ESCs and mesenchymal stem cells [201,202,203]. Interestingly, the catalysis of H2BK120ub was particularly important for transcriptional induction of relatively long genes during ESC differentiation [202]. Manipulating H2BK120ub levels by the silencing of *Rnf20*, H2BK120-deubiquitinase *Usp44*, or the ectopic expression of mutant H2BK120R inhibits ESC differentiation, emphasizing the importance of H2BK120ub in this process [202,203]. Furthermore, it was shown that RNF40 was specifically required for the catalysis of H2BK120ub during the reprogramming of somatic cells into induced pluripotent stem cells (iPSCs), and that deletion of *Rnf40* impaired early gene activation and iPSC generation [204]. Similarly, RNF20 was highly expressed at early stages of somatic cells reprogramming, and its knockout resulted in reprogramming failure [205]. Taken together, these studies highlight the importance of H2BK120ub dynamics in ESC plasticity and differentiation.

One example of the involvement of H2B120ub in tissue-specific stem cell differentiation was shown in the production of astrocytes, a sub-type of glia cells differentiated from neural precursor cells [206]. Specifically, conditional deletion of *Rnf20* in the mouse brain impaired astrocyte differentiation, while the overexpression of *Rnf20* in primary neural precursor cells promoted their differentiation into astrocytes [196]. The link between H2BK120ub and glia cells was also demonstrated in Schwann cells, which represents the major type of glia cells in the periphery nervous system [207]. Molecular dissection indicated that RNF40 cooperates with early growth response 2 (EGR2) to regulate Schwann cell maturation. RNF40 physically interacts with EGR2, a transcription factor critical for peripheral nerve myelination [208,209], and the distribution of H2BK120ub deposition overlaps with EGR2 genomic binding, including in genes related to myelin proteins and lipid biosynthesis involved in myelin sheath formation [192]. The conditional deletion of *Rnf40* in Schwann cells resulted in impaired postnatal induction of Schwan cells terminal differentiation, reduced myelin production, and progressive axonal degradation. Mice lacking *Rnf40* in Schwann cells exhibited progressive gait abnormalities, ataxia, tremor, and hindlimb clasping during early postnatal stages, and by postnatal day 56 they showed severe movement problems and did not reach normal age [192].

Another example is the case of osteoblast differentiation, where RNF40 was required for the induction of specific differentiation markers and transcriptional reprogramming during early stages of human mesenchymal stem cells differentiation in vitro [201] and mouse in vivo studies [197]. Osteoblast-specific deletion of *Rnf40* in mice during early osteoblast differentiation stages resulted in defects in differentiation and decreased osteocytes number and function, leading to impaired bone formation [197]. Interestingly, the deletion of *Rnf40* during later stages of osteoblast differentiation using *Bglap*-driven expression of Cre recombinase did not affect osteocytes number and no bone formation defects were observed, although some increase in cortical and trabecular bone was detected. The deletion of *Rnf40* in late-stage osteoblast and osteocytes, however, did not result in gross changes or bone mass [197]. Taken together, these indicate the stage-dependent requirement for RNF40 activity in osteoblast differentiation and bone formation.

Finally, RNF20 and RNF40 are also involved in heart development. Using an in vivo forward genetics screen, both *Rnf20* and *Rnf40* genes were identified as regulators of cardiomyocytes maturation. Disruption of RNF20 or RNF40 expression impaired cardiomyocyte maturation in young mouse neonates, causing cardiac dysfunction. RNF20 was also required for maintenance of adult cardiomyocytes maturity features [121]. Furthermore, RNF20/40 complex and H2BK120ub were shown to affect cilia motility and regulate heart development in *Xenopus* and mice [210]. Human genetic studies revealed a contribution of mutations in RNF20/40 to congenital heart disease, which is a born structural abnormality of the heart [210,211]. On the other hand, RNF20 chromatin localization and H2BK120ub deposition were found to be lost in the differentiated state of muscle cells, [212] although the exact mechanism involving RNF20 and H2BK120ub during myogenic differentiation is still unclear.

### 4.2. Roles of RNF20 and RNF40 in Tissue Homeostasis and Human Malignancy

As reviewed extensively in the literature, there is a tight connection between the activity of RNF20/40 and H2BK120ub and cancer development [47,115,149,213]. Decreased H2BK120ub levels have been shown in various malignant cancer types, including breast, colorectal, lung, and parathyroid [193,213,214,215], suggesting that H2BK120ub and related factors may be used as cancer indicators and novel targets for therapy. Intriguingly, however, RNF20 and RNF40 levels were correlated with both promotion and suppression of tumorigenesis. For example, RNF20-depleted human MCF10A non-malignant epithelial cells and mouse NIH-3T3 fibroblasts showed higher oncogenic potential in various oncogenic cell behavior tests, suggesting that RNF20 is a tumor suppressor [150], while in laminal breast cancer cells RNF20 silencing reduced proliferation, migration, and tumorigenic and metastatic capacity, indicating its tumor promoting role [216].

The mechanism of tumor suppression by RNF20/40 may be through its association with tumor suppressor genes, such as the p53 tumor suppressor gene, involved in DDR and cell cycle arrest. Specifically, the RNF20/40 complex was shown to interact with p53 and regulate p53 target genes, while depletion of RNF20 decrease the expression of canonical p53 target genes [29,150,191,217]. Cell division cycle 73 (CDC73), which is a member of the PAF1 complex, is another tumor suppressor that directly interacts with RNF20/40 and is required for maintenance of H2BK120ub levels [215]. Another mechanism may be through genome instability associated with double strand breaks (DSBs) [218]. RNF20 is localized at sites of DSBs and is involved in DSB-induced H2B ubiquitination local chromatin organization, so that DNA repair proteins such as RAD51 and BRCA1 can be recruited to sites of DSBs [117]. Likewise, RNF40 is also required for the repair of DSBs, and impairs cell cycle checkpoint activation [145]. Furthermore, RNF20/40-deficient mouse fibrosarcoma cells showed compromised genome integrity and increased DSBs that evolved to specific genomic rearrangements, which may induce formation of an invasive phenotype [219].

In addition to the extensive cancer development research using mainly in vitro methods, there are several important studies that used mouse models to determine the roles of RNF20/40 in this process. We hereby focus on published work including heterozygous mice or conditional deletion in mice.

Tumor initiation and progression may result in some cases from chronic inflammation. RNF20 or RNF40 knockdown in human mammary MCF10A epithelial cells was shown to downregulate non-inflammatory genes and upregulated mature cytokine mRNA transcripts. Moreover, heterozygous *Rnf20*^+/−^ mice showed increased proinflammatory transcriptional response, and were shown to be more prone to severe colonic inflammation compared to wild-type mice following dextran sodium sulfate (DSS) treatment [193]. More immune cells were observed in *Rnf20*^+/−^ under these conditions, and their T cell functionality was reduced. Colons of *Rnf20*^+/−^ mice showed compromised barrier function and were shorter. Treatment with a chemical carcinogen and DSS resulted in more colorectal polyps in *Rnf20^+/−^* mice, which often progressed to inflammation-associated colorectal cancer [193]. In correspondence with these results, samples from human patients with ulcerative colitis, Crohn’s disease, and colorectal tumors, showed downregulation of RNF20/40 and H2BK120ub in both epithelium and stroma [193,200]. Similarly, when *Rnf20* or *Rnf40* were conditionally deleted in the mouse intestine, spontaneous colorectal inflammation occurred, and these knockout mice displayed significantly decreased survival [200]. Colon length was also decreased in these mice, and their colons showed severe epithelial damage and lymphoid infiltrates, while heterozygous *Rnf20* or *Rnf40* mice did not show similar phenotypes. Furthermore, RNA-seq results showed that the deletion of *Rnf20* or *Rnf40* promoted inflammatory bowel disease-related gene expression programs [200]. In contrast, DSS treatment given to mouse models with colon-specific heterozygous or homozygous deletion of *Rnf40,* resulted in lower weight loss, milder epithelial damage, and reduced colon inflammation when compared to control mice, indicating that RNF40 activity promotes inflammatory pathways in the mouse colon [199].

A specific connection between RNF20/40 and cancer was also shown from studies of human breast cancer patient’s samples, as well as from in vivo experiments [115,150,216,220]. Specifically, epithelial-specific deletion of *Rnf40*, either heterozygous or homozygous, in a mammary carcinoma mouse model increased their tumor-free survival and decreased the number of tumors they developed along with the kinetics of tumor growth. Complete deletion of *Rnf40* showed better results than partial deletion in this case, and no marked morphological changes were observed [198]. Taken together, these studies indicate that the effect of RNF20/40 on tumorigenesis potential is context-dependent.

## 5. Concluding Remarks

Tissue development and homeostasis in multicellular organisms relies on the ability to activate distinct sets of transcriptional programs [1,221,222]. A deep understanding of the exact mechanisms controlling these processes in a cell type-dependent and tissue-specific manner remains one of the major challenges in modern biology. Here, we focused on the two major forms of histone mono-ubiquitination involved in transcriptional regulation, H2AK119ub and H2BK120ub, with an emphasis on their roles in tissue development and disease.

While H2A119ub is generally associated with gene repression, H2BK120ub is mostly associated with transcriptional activation and elongation. Their opposing roles may be linked to the recognition by different readers, as well as their localization on opposing edges of the nucleosome [42,223,224]. Nonetheless, both H2AK119ub and H2BK120ub have emerged as critical regulators of stem cell identity, tissue development, and homeostasis, in many tissues and cell types [61,154,155]. Intriguingly, a repeating theme which emerges from different studies is that the role of each of these histone mono-ubiquitination events seem to be context-dependent and can either promote stemness or differentiation programs. Similarly, the deposition of H2AK119ub or H2BK120 exerts either tumor suppressive or oncogenic functions in different tissues and cell types [60,104,115,149,169,216]. Recent advances in sequencing-based platforms using low cell number inputs [225,226,227] now enable researchers to get new insights into the in vivo genomic localization, epigenetic landscape, and the dynamic changes of H2AK119ub and H2BK120ub in both rare and abundant stem cell populations. Such techniques hence provide a promising avenue to deepen our understanding of the molecular mechanisms controlling gene expression.

The use of genetic mouse models has also proven to be instrumental in elucidating the roles of PRC1-mediated H2AK119ub, as well as RNF20/40-mediated H2BK120ub, in somatic stem cells and their differentiated progenies (Table 1 and Table 2). Further dissection of their roles in additional tissues and cell types is expected to not only close the gaps in our understanding of the transcriptional regulatory mechanisms controlling tissue development, but also provide a better understanding of the mechanisms that go awry in human pathologies.

## Figures and Tables

**Figure 1 cells-11-02404-f001:**
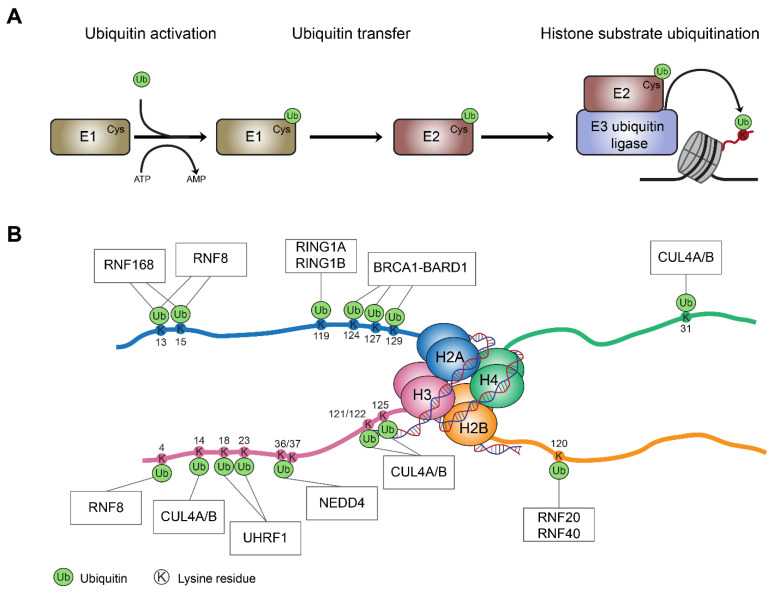
E3 ubiquitin ligases and histone ubiquitination events. (**A**) The process of histone mono-ubiquitination. Ubiquitin is first attached to E1 enzyme and activated in an ATP-dependent manner. Next, activated ubiquitin is transferred to E2 enzyme. Finally, E2 interacts with an E3 ubiquitin ligase that covalently attaches the ubiquitin to a target lysine residue in the histone. (**B**) A schematic representation of histone lysine (K) residues that undergo ubiquitination (Ub), and their corresponding E3 ubiquitin ligases. Mono-ubiquitination at K13 and K15 on histone H2A (H2AK13ub and H2AK15ub) placed by RNF168 are associated with DDR and DNA replication. Notably, K13 and K15 on histone H2A and K4 on histone H3 are targets for poly-ubiquitination by RNF8. Mono-ubiquitination at K119 on histone H2A (H2AK119ub) is written by RING1A/B members of the Polycomb-repressive complex 1 (PRC1). The mono-ubiquitination of K124, K127, and K129 on histone H2A (H2AK124ub, H2AK127ub, and H2AK129ub) involved in DDR, is catalyzed by the BRCA1-BARD1 complex. The mono-ubiquitination of K18 and K23 on histone H3 (H3K18ub and H3K23ub) written by UHRF1 is associated with the maintenance of DNA methylation. Mono-ubiquitination of K14, K121, K122, and K125 on histone H3 (H3K14ub, H3K121ub, H3K122ub, and H3K125ub) is catalyzed by CUL4A/B, and is involved in regulation of chromatin structure, while CUL4A/B-mediated mono-ubiquitination of K31 on histone H4 (H4K31ub) is associated with DDR. Finally, mono-ubiquitination of K120 on histone H2B by the RNF20/40 complex is associated with transcriptional elongation.

**Figure 2 cells-11-02404-f002:**
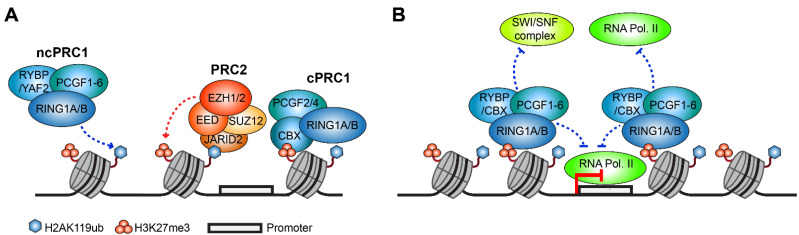
Transcriptional regulation by PRC1-mediated H2AK119ub. (**A**) Polycomb complexes are divided into two principal groups: canonical PRC1 (cPRC1) and non-canonical PRC1 (ncPRC1). ncPRC1 complexes are proficient for H2AK119ub catalysis, and their recruitment is independent of PRC2. The deposition of H2AK119ub promotes the recruitment of PRC2 via its JARID2 subunit. cPRC1 complexes are recruited in a PRC2/H3K27me3-dependent manner via their CBX subunit and possess low H2AK119ub catalytic activity. Instead, cPRC1 complexes synergize with PRC2 to compact chromatin and establish transcriptionally silent Polycomb domains. (**B**) Schematic illustration of H2AK119ub direct role in transcriptional repression. The deposition of H2AK119ub may affect transcription at multiple levels, by limiting accessibility of chromatin remodeling complexes and the transcription initiation machinery, as well as by blocking and interfering with RNA Polymerase II activity.

**Figure 3 cells-11-02404-f003:**
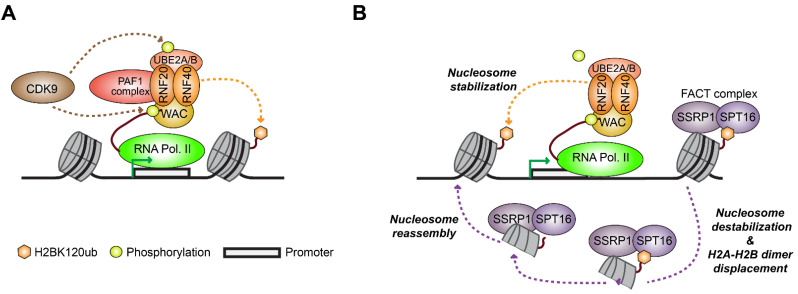
RNF20/40 and H2BK120ub facilitates transcriptional elongation. (**A**) To initiate H2B ubiquitination, RNF20/40, an E2 enzyme (UBE2A/B), and PAF1 create a complex that links to RNA polymerase II (Pol II) through WAC, following CDK9-dependent RNA Pol II phosphorylation. (**B**) H2BK120ub promotes FACT accumulation via interactions with the Spt16 subunit of FACT complex. In turn, the activity of FACT enables RNA Pol II transcriptional elongation by displacing the H2A/H2B dimer from the nucleosome. Subsequently, H2BK120ub and Spt16 also cooperate to promotes nucleosome reassembly and stabilization.

**Table 1 cells-11-02404-t001:** Genetic mouse models for catalytic-inactive PRC1 complexes.

Ligase	Mouse Model Used	Tissue	Phenotypes Observed	Reference
RING1B	*Ring1b* ^I53A/I53A^	Germline mutation	Perinatal lethality; delayed/impaired embryonic development; edema; exencephaly.	[173]
RING1A, RING1B	*K14-Cre*; *Ring1a*^−/−^*Ring1b*^flox/I53A^	Embryonic skin epithelium	Ectopic formation and increased Merkel cell numbers in newborn mouse back skin.	[90]
RING1A, RING1B	*K14-Cre*^ERT2^; *Ring1a*^−/−^*Ring1b*^flox/I53A^	Adult skin epidermis	Increase in epidermal melanocytes and skin pigmentation.	[174]

**Table 2 cells-11-02404-t002:** Genetic mouse models of RNF20 and RNF40.

Ligase	Mouse Model Used	Tissue	Phenotypes Observed	References
RNF20	*Stra8-Cre*; *Rnf20*^flox/flox^	Testis	Impaired spermatogenesis, and male infertility.	[195]
RNF20	*GFAP-Cre*; *Rnf20*^flox/flox^	Brain	Decreased astrocyte production during brain development.	[196]
RNF40	*Dhh-Cre*; *Rnf40*^flox/flox^	Peripheral nervous system	Sever hypomyelination and axonal degeneration.	[192]
RNF40	*Runx2-Cre*; *Rnf40*^flox/flox^	Bone	Impaired osteoblast differentiation into mature osteocytes, reduced bone formation and matrix apposition rates, and increased bone mass.	[197]
RNF40	*Bglap-Cre*; *Rnf40*^flox/flox^	Bone	No change in osteocytes, but decreased number of osteoclasts, and increased cortical and trabecular bone mass.	[197]
RNF40	*Dmp1-Cre*; *Rnf40*^flox/flox^	Bone	No change in osteoblast differentiation, or bone mass.	[197]
RNF20	*R26-Cas9*; CASAAV-*Rnf20* and *AAV*-*TnT-Cre*; *Rnf20*^flox/flox^		Immature cardiomyocytes phenotype in neonatal mice, and disrupted T-tubule organization in adult mice.	[121]
RNF20	*Rnf*20^+/−^	Germline deletion	Mice were prone to severe colonic inflammation.	[193]
	*MMTV-Erbb2*; *MMTV-Cre*; *Rnf40*^flox/flox^	Mammary gland	Significant decrease in breast tumor development and increased survival.	[198]
RNF40	*CAC-Cre; Rnf40^flox/flox ^* and *CAC-Cre; Rnf40^wt/flox^*	Colon	Reduced inflammation, weight loss and epithelial damage.	[199]
RNF20/40	*Villin-Cre*^ERT2^; *Rnf20*^flox/flox^ and *Villin-Cre*^ERT2^; *Rnf40*^flox/flox^ and *Villin-Cre*^ERT2^; *Rnf20*^flox/flox^; *Rnf40*^flox/flox^	Intestine	Spontaneous colorectal inflammation in adult mice, reduced survival, decreased colon length, severe epithelial damage, increased vascularization.	[200]

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
