# Peer review of "Histone Mono-Ubiquitination in Transcriptional Regulation and Its Mark on Life: Emerging Roles in Tissue Development and Disease"

_cells, 2022, doi:10.3390/cells11152404_

Round 1

Reviewer 1 Report

The manuscript by Oss-Ronen et al. provides a focused review on the biological role of two histone mono-ubiquitinations and its contribution to transcriptional control in tissue development and disease. The authors focus on two specific marks, the mono-ubiquitination of histone 2A at lysine 119 (H2AK119ub) and histone H2B at lysine 120 (H2BK120ub). The authors describe the complexes depositing these modifications and then summarise the role of both marks in transcriptional regulation, followed by their involvement in tissue development, cell differentiation and pathologies (e.g. cancer).

The manuscript is well-structured, understandable and provides a comprehensive and useful overview of the field. The figures are well-designed and enhance understanding of the text.

Below, I provided a few comments on how to improve the manuscript.

Major comments:

1) Several parts of the manuscript are quite specialised and may not be readily accessible to a non-expert reader:

-> l.66-77: the part describing the canonical and non-canonical PCR1 sub-complexes is quite complex and may be difficult to understand for a non-expert, it may be useful to include a drawing of the complexes’ subunits and indicate their function.

-> l. 178-180: “Indeed, in addition to the catalysis of H2AK119ub, cPRC1 complexes can exert gene repression by promoting phase separation, and high-order chromatin organization and nuclear sub-clustering”: this sentence is quite specialised, it may be useful to rephrase/expand for non-expert readers what the authors mean by phase separation and nuclear sub-clustering.

-> l. 189-191: “fertilization and preimplantation stages, maternally inherited H2AK119ub at canonical Polycomb target genes is being equalized at the paternal allele starting the two-cell stage, while being depleted from atypical targets and gene deserts”: it is not clear what equalized means in this context (deposited at the paternal allele?), also “at” should probably be added before two-cell stage. This part would benefit from some expansion: How is the allele-specific deposition achieved? What are the the typical and atypical targets mentioned?

-> “The mechanism of tumor suppression by RNF20/40 may be through its association with tumor suppressor promoters, such as the protein p53, and cell division cycle 73 (CDC73), which is a member of the PAF1 complex. Another mechanism may be through genome instability associated with DSBs. RNF20/40-deficient cells showed compromised genome integrity and increased DSBs that evolved to specific genomic rearrangements, which may induce formation of an invasive phenotype”. Could the authors expand on the mechanisms here? What is the mechanistic connection between the mono-ubiquitination and tumor suppression and between mono-ubiquitination and double strand breaks?

2) The biological roles of the modifications are comprehensively reviewed and illustrated with examples, but the mechanistic link between the modifications and the biological outcome is not provided. It would be useful to include a paragraph/section on how both mono-ubiquitinated histones are recognised and read by other proteins to confer the biological meaning. Are there any specific conserved ubiquitin reading domains that recognise the mark? Is there any structural data explaining how the specific recognition of the modifications is achieved?

3) What is known about the genomic distribution of 2AK119ub and H2BK120ub across functional elements?

Minor comments:

1) l.84-93: the order of events is not entirely clear here. How is cPCR1 recruited? If it is recruited by PRC2-mediated H3K27me3, what complex deposits H2AK119ub in the first place?

2) some language errors and unclear formulations are present:

-> l.10: “have emerged” should be changed to “has emerged”

-> l.36: “(E2) conjugates the ubiquitin via a thioester bond to a cystine residue”:  it is unclear what the cytosine residue refers to

-> l.153-154: “Finally, RNF20/40-mediated H2BK120ub is also associated with the activity of the facilitates chromatin transcription (FACT) complex”: the sentence is somehow cut (after the activity of the…)

-> Abbreviation should be explained when first time introduced (e.g. EGR2, DSBs etc)

Author Response

The manuscript by Oss-Ronen et al. provides a focused review on the biological role of two histone mono-ubiquitinations and its contribution to transcriptional control in tissue development and disease. The authors focus on two specific marks, the mono-ubiquitination of histone 2A at lysine 119 (H2AK119ub) and histone H2B at lysine 120 (H2BK120ub). The authors describe the complexes depositing these modifications and then summarise the role of both marks in transcriptional regulation, followed by their involvement in tissue development, cell differentiation and pathologies (e.g. cancer).

The manuscript is well-structured, understandable and provides a comprehensive and useful overview of the field. The figures are well-designed and enhance understanding of the text.

We thank the reviewer for their strong support of our review article and for the excellent suggestions to further improve this review. The revised manuscript now addresses the comments made by the reviewer, as described below.

Below, I provided a few comments on how to improve the manuscript.

Major comments:

1) Several parts of the manuscript are quite specialised and may not be readily accessible to a non-expert reader:

-> l.66-77: the part describing the canonical and non-canonical PCR1 sub-complexes is quite complex and may be difficult to understand for a non-expert, it may be useful to include a drawing of the complexes’ subunits and indicate their function.

This is a great suggestion made by the reviewer. We have modified Figure 2A illustration to include both canonical PRC1 (cPRC1) and non-canonical PRC1 (ncPRC1) complexes, and provided in the figure legends an explanation of the key differences between the two main groups of PRC1 sub-complexes.

-> l. 178-180: “Indeed, in addition to the catalysis of H2AK119ub, cPRC1 complexes can exert gene repression by promoting phase separation, and high-order chromatin organization and nuclear sub-clustering”: this sentence is quite specialised, it may be useful to rephrase/expand for non-expert readers what the authors mean by phase separation and nuclear sub-clustering.

We thank the reviewer for pointing to this potential difficulty. To clarify this sentence for the non-expert readers, we have added the following sentence, which describes the effect of phase separation and nuclear sub-clustering on chromatin and transcription: “Indeed, in addition to the catalysis of H2AK119ub, cPRC1 complexes can exert gene repression by promoting phase separation and high-order chromatin organization and nuclear sub-clustering, leading to a compacted and transcriptionally silent chromatin” (Chapter 3, page 7, lines 275-278).

-> l. 189-191: “fertilization and preimplantation stages, maternally inherited H2AK119ub at canonical Polycomb target genes is being equalized at the paternal allele starting the two-cell stage, while being depleted from atypical targets and gene deserts”: it is not clear what equalized means in this context (deposited at the paternal allele?), also “at” should probably be added before two-cell stage. This part would benefit from some expansion: How is the allele-specific deposition achieved? What are the the typical and atypical targets mentioned?

We have rephrased this sentence, which now describes what are the canonical Polycomb targets, as well as the meaning of the term equalized. The text now reads as follows: “…maternally inherited H2AK119ub at canonical Polycomb target genes (developmental genes enriched for PRC1/2 binding in mouse ESCs) is being equalized and reaches to similar high levels in the paternal allele at the two-cell stage, while being depleted from atypical targets and gene deserts” (Chapter 3.1, page 7, lines 287-292).

-> “The mechanism of tumor suppression by RNF20/40 may be through its association with tumor suppressor promoters, such as the protein p53, and cell division cycle 73 (CDC73), which is a member of the PAF1 complex. Another mechanism may be through genome instability associated with DSBs. RNF20/40-deficient cells showed compromised genome integrity and increased DSBs that evolved to specific genomic rearrangements, which may induce formation of an invasive phenotype”. Could the authors expand on the mechanisms here? What is the mechanistic connection between the mono-ubiquitination and tumor suppression and between mono-ubiquitination and double strand breaks?

We have expanded the explanation about the connection between mono-ubiquitination and the function of the tumor suppressors p53 and CDC73 in chapter 4.2 in the revised manuscript (Page 11, lines 444-450). The revised text also provides more details regarding the connection between RNF20/40-mediated H2BK120ub and DSBs (Page 11, lines 450-458).

2) The biological roles of the modifications are comprehensively reviewed and illustrated with examples, but the mechanistic link between the modifications and the biological outcome is not provided. It would be useful to include a paragraph/section on how both mono-ubiquitinated histones are recognised and read by other proteins to confer the biological meaning. Are there any specific conserved ubiquitin reading domains that recognise the mark? Is there any structural data explaining how the specific recognition of the modifications is achieved?

Following the reviewer's important comment, we now included in the revised manuscript examples regarding the recognition of the ubiquitinated histone by different specific “readers” of the ubiquitin mark, as well as on the epigenetic crosstalk between H2AK119ub/H2BK120ub and other PTMs (see chapter 2.1 page 3 lines 120-123; page 4 lines 137-141; page 4 lines 145-147; page 4 lines 163-165, and chapter 2.2 page 6 lines 223-235).

3) What is known about the genomic distribution of 2AK119ub and H2BK120ub across functional elements?

H2AK119ub is mostly enriched in the promoter regions, while H2BK120ub is mostly present at gene bodies of actively transcribed genes. This information is now included in the revised manuscript in page 3 lines 100-102, and page 5 lines 192-194, respectively.

Minor comments:

1) l.84-93: the order of events is not entirely clear here. How is cPCR1 recruited? If it is recruited by PRC2-mediated H3K27me3, what complex deposits H2AK119ub in the first place?

We thank the reviewer for pointing out this unclarity. cPRC1 complexes are recruited via the recognition of the CBX subunit to PRC2-mediated H3K27me3. This information is included in chapter 2.1, page 3, lines 108-111. In addition, we have revised Figure 2A, and included this information in the illustration, making a clear separation between cPRC1 and ncPRC1. Regarding ncPRC1, their recruitment is independent of PRC2/H3K27me3 and in fact drives the recruitment of PRC2 complexes. Expansion regarding the recruitment of ncPRC1 is included in the revised text in chapter 2.1, page 3, lines 113-117, and now reads as follows: “The recruitment of ncPRC1 complexes is not dependent of PRC2/H3K27me3, and instead can be mediated by the KDM2B subunit of ncPRC1.1 in a generic fashion to non-methylated DNA sequences enriched for CpG islands, or can be directed to sequence-specific loci via the interaction of ncPRC1 complexes with accessory transcription factors”.

2) some language errors and unclear formulations are present:

-> l.10: “have emerged” should be changed to “has emerged”

The grammatical error was corrected in the revised manuscript, page 1, line 11.

-> l.36: “(E2) conjugates the ubiquitin via a thioester bond to a cystine residue”:  it is unclear what the cytosine residue refers to

The sentence has been rephrased in the revised manuscript, to clarify the sequence of events, and now reads as follow: "… then a cystine residue on a ubiquitin-conjugating enzyme (E2) is conjugated to the ubiquitin via a thioester bond; and finally, RING finger ubiquitin-ligases (E3) transfer the ubiquitin from E2 to a target lysine residue…" (page 1, lines 36-39).

-> l.153-154: “Finally, RNF20/40-mediated H2BK120ub is also associated with the activity of the facilitates chromatin transcription (FACT) complex”: the sentence is somehow cut (after the activity of the…)

We expanded this sentence in the revised manuscript, and it now reads as follows: " RNF20/40-mediated H2BK120ub is also associated with the activity of the FACT complex in nucleosome disassembly and reassembly during transcription." (Chapter page 6, lines 248-251).

-> Abbreviation should be explained when first time introduced (e.g. EGR2, DSBs etc)

We added full names in the text to all of the abbreviations that required explanation, including the examples that were indicated by the reviewer. All of the new abbreviations added also appear under the “Abbreviations” section.   

Reviewer 2 Report

General comments:

Overall Oss-Ronen et al. have done a good job of  covering recent studies of histone monobiquitination in development and adult tissue.

1)    The review is focused exclusively on PRC complex and RNF 20/40. It would be nice to have a few sentences on other monoubiquitinating enzymes, pre-focus.

2)    Deubiquitination is mentioned in a half sentence. It would be nice to have a short paragraph on regulation of PRC, RNF 20/40 in terms of what stimuli activate them to add monoUb to histones, as well as a bit more on how various deubiquitinases (SAGA?) fit into the regulation/inactivation of histone monoubiquitinatioin.

3)    Are there any other types of ubiquitin linkages relevant for histone modification?  Maybe a word or two? Polyubiquitination?

4)    Does crosstalk between Ub-histone and other PTMs affect tissue homeostasis?

Specific comments:

1)    Line 99. “may repress gene expression by interfering with transcriptional machinery…”. Monoubiquitin is  a small molecule, so steric hindrance is less likely. Please clarify.

2)    Line 125. Please clarify why activity of the RNF20/40 complex depends on E2s (vs E3).

Author Response

Overall Oss-Ronen et al. have done a good job of  covering recent studies of histone monobiquitination in development and adult tissue.

We thank the reviewer for their support of our review article, and specifically for their great suggestions to include a broader view of various aspects related to histone ubiquitination. The revised manuscript now addresses the comments made by the reviewer, as described below.

1)    The review is focused exclusively on PRC complex and RNF 20/40. It would be nice to have a few sentences on other monoubiquitinating enzymes, pre-focus.

Following the reviewer's excellent suggestion, we have included a pre-focus on mono-ubiquitinating enzymes and the main cellular processes they are involved in. This info in included in the revised manuscript in pages 1-2, lines 41-47.

2)    Deubiquitination is mentioned in a half sentence. It would be nice to have a short paragraph on regulation of PRC, RNF 20/40 in terms of what stimuli activate them to add monoUb to histones, as well as a bit more on how various deubiquitinases (SAGA?) fit into the regulation/inactivation of histone monoubiquitinatioin.

We thank the reviewer for this great suggestion. The involvement of deubiquitination is now mentioned in the introduction (Page 2, lines 55-60), as well as more specifically when discussing PRC1 (Pages 3-4, lines 128-147), and RNF20/40 complex (Page 6, lines 232-248). The various mechanisms that promote mono-ubiquitination and spreading of PRC1 (Chapter 2.1, page 3, lines 120-123), and the catalytic activity of RNF20/40 (Chapter 2.2, page 5, lines 201-208; and Figure 3A) are also discussed to some extent in the revised manuscript.

3)    Are there any other types of ubiquitin linkages relevant for histone modification?  Maybe a word or two? Polyubiquitination?

The existence of poly-ubiquitination is now briefly mentioned in the introduction of the revised text, when describing RNF8 (see page 2, lines 47-49).

4)    Does crosstalk between Ub-histone and other PTMs affect tissue homeostasis?

We have included a short paragraph in the introduction regarding the concept of crosstalk between histone mono-ubiquitination and other PTMs (Page 2, lines 60-64; page 6, lines 223-235), but feel that further expansion on this crosstalk is outside of the main focus of our review paper and should be addressed elsewhere.

Specific comments:

1)    Line 99. “may repress gene expression by interfering with transcriptional machinery…”. Monoubiquitin is  a small molecule, so steric hindrance is less likely. Please clarify.

This is a very interesting point raised by the reviewer. Some of the involvement of H2AK119ub in repression is through the stabilization of PRC1 and PRC2 complexes, that can read the H2AK119ub mark, and their physical presence interrupts with transcription. This info is now included in page 4, lines 163-165 of the revised manuscript. That being said, the transcriptional repression of many genes is maintained in the absence of PRC2 complexes, and relies on the catalysis activity of PRC1 – as demonstrated using catalytic-inactive PRC1 mutant models (PMID: 31883950; 29727681; 32196702), suggesting additional mechanisms by which H2AK119ub interferes with the transcriptional machinery.  However, the association between H2AK119ub with the transcriptional machinery, and whether they directly interact, is poorly understood, as discussed in the manuscript text at the end of the paragraph (Page 4, lines 165-168).  

2)    Line 125. Please clarify why activity of the RNF20/40 complex depends on E2s (vs E3).

The RNF20/40 complex functions as an E3 ligase that catalyzes mono-ubiquitination by transferring the ubiquitin from the E2 enzyme to the substrate. Thus, activity of this complex depends on a specific E2 enzyme. We elaborated on this connection in the revised manuscript in page 5, lines 194-197. We also clarified the steps of the enzymatic ubiquitination reaction involving E1, E2 and E3 in the introduction (Chapter 1, page 1, lines 35-39), and added Figure 1A

Reviewer 3 Report

In this review article "Histone mono-ubiquitination and its mark on life: emerging 2 roles in tissue development and disease" authors are explained very nicely about the histone mono- ubiquitination. I would like to suggest few things that should be improve the impact of the article.

1. Authors can represent the enzymatic reaction of ubiquitination in the form of figure (line 34 to 38).

2. Elaborate the legend of the figure 1 with the proper abbreviation of the different residues

Author Response

In this review article "Histone mono-ubiquitination and its mark on life: emerging 2 roles in tissue development and disease" authors are explained very nicely about the histone mono- ubiquitination. I would like to suggest few things that should be improve the impact of the article.

We thank the reviewer for their support in our manuscript, and for their suggestions, which have been addressed as described below.

  1. Authors can represent the enzymatic reaction of ubiquitination in the form of figure (line 34 to 38).

We thank the Reviewer for this useful suggestion, and have updated the figure to include the enzymatic reaction involving E1, E2 and E3 enzymes, in a form of a figure. This is now shown in the new Figure 1A.

  1. Elaborate the legend of the figure 1 with the proper abbreviation of the different residues

Following the reviewer's suggestion, Figure 1B in the revised manuscript now includes elaboration of the abbreviations of the different lysine residues modified by ubiquitination

Reviewer 4 Report

This is a very nicely written review that completely summarized up-to-date understanding of Histone mono-ubiquitination and their roles in tissue development and disease. The discussion is adequate and the presentation of the data is appropriate and well balanced.

There are some minor points or suggestions that the reviewer wants to reminder the authors:

1, Some of the data should accurate, for instance, page 1, line 43, the occurrence rate of up to 15% and 1-2%, are they total ubiquitination or mono-ubiquitination? 

2, Some of the citations should be the original research articles instead of citing a review paper, especially the key discoveries or special data cited.

3, Page 5 line 188, "7-days old females", in what animals?  The descriptions about specific terms should have more details, like what cells, what tissues, what animals.

4, The title of this review is “Histone mono-ubiquitination”, however, throughout the manuscript, the authors mainly discussed H2AK119ub, H2BK120ub and their ubiquitin ligase PRC1 and RNF20/40. In Figure 1, there listed so many ubiquitination and their ligases for histone, is the title a little bit over stated?

Author Response

This is a very nicely written review that completely summarized up-to-date understanding of Histone mono-ubiquitination and their roles in tissue development and disease. The discussion is adequate and the presentation of the data is appropriate and well balanced.

We thank the reviewer for their support and appreciation of our review article, as well as for the important suggestions to further improve this review. The revised manuscript now addresses the comments made by the reviewer, as described below.

There are some minor points or suggestions that the reviewer wants to reminder the authors:

1, Some of the data should accurate, for instance, page 1, line 43, the occurrence rate of up to 15% and 1-2%, are they total ubiquitination or mono-ubiquitination? 

The occurrence rate mentioned in the text referred to total histone ubiquitination, and we have now clarified this point in the revised manuscript (Chapter 1, page 2, lines 52-54).

2, Some of the citations should be the original research articles instead of citing a review paper, especially the key discoveries or special data cited.

We have revised the manuscript and included citations of the original research articles with key discoveries when first describing PRC1 and RNF20/40 complexes.

3, Page 5 line 188, "7-days old females", in what animals?  The descriptions about specific terms should have more details, like what cells, what tissues, what animals.

The 7-days old females mentioned in the text referred to mice, and we added this information to the revised manuscript (Chapter 3.1, page 7, line 286). We have also revised the manuscript and added relevant details about cell types and animals used in the different studies described throughout the manuscript in places where such info was missing (for example see chapter 4.1, page 11, lines 439, 440, 456, and 464). 

4, The title of this review is “Histone mono-ubiquitination”, however, throughout the manuscript, the authors mainly discussed H2AK119ub, H2BK120ub and their ubiquitin ligase PRC1 and RNF20/40. In Figure 1, there listed so many ubiquitination and their ligases for histone, is the title a little bit over stated?

We accepted this remark and changed the title of the revised manuscript, to clarify that we reviewed specifically mono-ubiquitination in transcriptional regulation, which is the main role of H2AK119ub and H2BK120ub. The new title is: "Histone mono-ubiquitination in transcriptional regulation and its mark on life: emerging roles in tissue development and disease"

Round 2

Reviewer 1 Report

The authors have addressed all my comments and the manuscript has been improved for overall clarity and content. I therefore I recommend the acceptance of the manuscript.

Reviewer 3 Report

This manuscript is in the good shape.

Accepted.

Reviewer 4 Report

The authors addressed all my questions. It is suitable for publication at this time!